# Reliability of Running Stability during Treadmill and Overground Running

**DOI:** 10.3390/s23010347

**Published:** 2022-12-29

**Authors:** Dominik Fohrmann, Daniel Hamacher, Alberto Sanchez-Alvarado, Wolfgang Potthast, Patrick Mai, Steffen Willwacher, Karsten Hollander

**Affiliations:** 1Institute of Interdisciplinary Exercise Science and Sports Medicine, Faculty of Medicine, MSH Medical School Hamburg, 20457 Hamburg, Germany; 2Institute of Biomechanics and Orthopedics, German Sport University Cologne, 50933 Cologne, Germany; 3Institute of Sports Science, Friedrich Schiller University Jena, 07749 Jena, Germany; 4Department of Sports and Exercise Medicine, Institute of Human Movement Science, University of Hamburg, 20148 Hamburg, Germany; 5Department of Mechanical and Process Engineering, Offenburg University of Applied Sciences, 77652 Offenburg, Germany

**Keywords:** locomotion, biomechanics, inertial sensor, nonlinear time-series analysis

## Abstract

Running stability is the ability to withstand naturally occurring minor perturbations during running. It is susceptible to external and internal running conditions such as footwear or fatigue. However, both its reliable measurability and the extent to which laboratory measurements reflect outdoor running remain unclear. This study aimed to evaluate the intra- and inter-day reliability of the running stability as well as the comparability of different laboratory and outdoor conditions. Competitive runners completed runs on a motorized treadmill in a research laboratory and overground both indoors and outdoors. Running stability was determined as the maximum short-term divergence exponent from the raw gyroscope signals of wearable sensors mounted to four different body locations (sternum, sacrum, tibia, and foot). Sacrum sensor measurements demonstrated the highest reliabilities (good to excellent; ICC = 0.85 to 0.91), while those of the tibia measurements showed the lowest (moderate to good; ICC = 0.55 to 0.89). Treadmill measurements depicted systematically lower values than both overground conditions for all sensor locations (relative bias = −9.8% to −2.9%). The two overground conditions, however, showed high agreement (relative bias = −0.3% to 0.5%; relative limits of agreement = 9.2% to 15.4%). Our results imply moderate to excellent reliability for both overground and treadmill running, which is the foundation of further research on running stability.

## 1. Introduction

Local dynamic stability describes the human motor control system’s ability to compensate for small perturbations during locomotion [1]. It can be determined as the largest Lyapunov exponent derived from any source of kinematic data [2,3]. For walking gait, local dynamic stability has been studied extensively [4,5,6,7] and has shown an association with the risk of falling [8,9]. 

In running research, local dynamic stability has been less frequently studied. Running stability may, however, provide important insights into runners’ movement abilities under different conditions. It has been shown that differences in running stability are associated with varying running speed [10,11], surface conditions [12], footwear [13,14], level of running experience [15,16], and stages of a maximum effort run [16]. 

For its practical and clinical relevance, the reliability of running stability must be evaluated. To date, the reliability of running stability has only been investigated using expensive opto-electrical measurements in a laboratory-based setting [17]. These findings showed good reliability both within days, between three consecutive days, and between two months (intraclass correlation coefficient > 0.7).

Using wearable inertial sensors provides the opportunity to determine running stability in the absence of an expensive multi-camera system and outside a laboratory setting. The reliability of inertial sensor-based measurements of the local dynamic stability was only evaluated for walking thus far [18,19,20,21]. They revealed mainly moderate-to-good reliability (intraclass correlation coefficient: 0.58–0.84). Results varied depending on the location of the sensors, the number and choice of signal axes, and the exact calculation method. Still, the reliability of running stability based on inertial sensor measurements has not been reported.

Precise and reliable measurements of the local dynamic stability require the analysis of a large number of consecutive strides [19,22]. Thus, most studies investigating running stability utilize motorized treadmills during data acquisition. Yet, it remains unclear how these findings translate to runners’ real-world situations. Most kinematic parameters are in good agreement when compared between treadmill and overground running [23]. However, this has not yet been shown for running stability. Dingwell et al. [24] found a small but significant reduction of local dynamic stability during treadmill walking compared to overground walking in young, healthy adults. However, it is not known whether this can be translated to running stability.

The primary purpose of this study was to estimate the test-retest reliability of inertial sensor-based measurements of the running stability at different body locations. Secondly, this study aimed to evaluate the agreement of running stability measured (a) on a treadmill, (b) overground in a laboratory-like setting, and (c) overground in an outdoor setting.

## 2. Materials and Methods

### 2.1. Participants

The study was approved by the University’s ethics committee (MSH-2022/184). Twenty-five participants were recruited through local sports clubs, social media advertisements, and word of mouth (participants referring to other eligible runners). Competitive runners and triathletes with an average weekly mileage of at least 20 km between the age of 18 and 65 years were included in the study. Participants were required to have participated in at least one official running event in the last year. Exclusion criteria were any lower-limb injuries in the three months before the study started. All participants gave written informed consent to participate in the study, in accordance with the Declaration of Helsinki. 

### 2.2. Data Collection

Participants visited the laboratory three times on two consecutive days between July and September 2022. They were measured twice on the first and once on the second day. The first and second visits were in the morning and afternoon, respectively, at least six hours apart. Half of the participants’ third visit was on the following day’s morning. The other half was during the afternoon. The participants had to refrain from any lower-limb exercises between twenty-four hours before the first visit and until after the last visit. The participants had to bring the same running shoes to every visit. 

During each visit, participants performed a self-paced warm-up on a motorized treadmill for 5 min (h/p/cosmos sports & medical GmbH, Nussdorf-Traunstein, Germany). Thereafter, participants ran under three conditions at their preferred running speed: On the treadmill at a constant speed for five minutes;Multiple laps across a 50 m secured corridor in the laboratory building (overground indoors);Multiple laps across a 50 m flat concrete walkway outside near the laboratory building (overground outdoors).

For both overground running conditions, the participants performed laps similar to shuttle-runs over a 50 m track. The running speed was predetermined through a sound signal and marks along the tracks 10 m apart. The participants had to adopt their running speed so they were at the next mark at the next sound signal. The correct speed was verified through opto-electrical light barriers over the middle 30 m (WittyGATE, Microgate Srl, Bolzano, Italy). Six laps (i.e., 300 m) were considered familiarization to the method and speed. Another ten laps (500 m) were then recorded for analysis. The condition order was randomized for each participant and visit (block-randomization; block size = 6). During all conditions, the participants ran at their preferred running speed. 

During the first visit, the participants’ preferred running speed was determined through the method of limits [25], as described previously [17]. In brief, participants ran on the treadmill starting at 7 km/h. An experienced lab engineer increased the treadmill speed every ten seconds by 0.5 km/h. The participants had no information on the treadmill speed. Participants were asked to give notice when a comfortable running speed was achieved. Upon reach, this speed was written down, and the treadmill was set to 3 km/h faster than this. After a short period, the speed was then decreased by 0.5 km/h every ten seconds. Again, participants were asked to give notice when a comfortable running speed was achieved. This procedure was repeated twice or until the denoted speeds differed by less than 10%. The preferred running speed was then considered the average of the last two trials. 

Three-dimensional angular velocity signals were captured using four time-synchronized inertial sensors (128 Hz, ±2000°/s gyroscope measurement range, OPAL, Apdm, Portland, OR, USA). They were securely mounted to the foot, tibial tuberosity, sacrum, and xiphoid process of the sternum. For this, a customized button-system was used which was integrated in the shoelaces, the seam of calf-compression garments, and tight running shorts and using an elastic chest strap. The same researcher (D.F.) placed the sensors at each measurement. 

### 2.3. Data Processing

Unfiltered three-dimensional gyroscope signals were processed using custom Python scripts (Python version 3.10, 2022, python.org (accessed on 7 March 2022)). We determined running stability from the maximum short-term divergence exponent λ_S_. The calculation of λ_S_ has been described in detail elsewhere [4]. In short, separate strides were identified through ipsilateral initial foot contacts. These were detected based on the minimum of the pitch-gyroscope signal of the foot-mounted sensor (k1 in [26]). For the treadmill trials, we chose the middle 100 strides of the five minutes. These 100 strides were time-normalized to a length of 10,000 samples. For both overground conditions, the middle ten strides of each of the ten measured laps were chosen. Thereby, the first and last strides were excluded to ensure quasi-stationarity. These bouts were then concatenated and time-normalized in the same manner, also resulting in signals of length 10,000 samples. 

From these signals, a state-space vector was reconstructed through time-delayed embedding [27] in the form of the following:
(1)$$s(t)=\left[x(t),x(t+\tau ),x(t+2\tau )\ldots ,x(t+(d_{E}-1)\tau )\right]$$
where *x(t)* is the time-normalized gyroscope signal, *τ* the time delay, and *d_E_* the embedding dimension. Time delays were determined using the first minimum of the average mutual information [28]. The integer mean over the three sensor axes and conditions was chosen for each sensor location (sternum: 11, sacrum: 6, tibia: 8, and foot: 10). The embedding dimension was calculated from the global false nearest neighbors algorithm [29]. Since three-dimensional signals were used, this could only result in a multiple of 3. For each condition, the maximum *d_E_* was chosen per sensor (sternum: 9, sacrum: 9, tibia: 6, and foot: 9). The average logarithmic divergence of each time point was computed using Rosenstein’s algorithm [30]. The resulting logarithmic divergence curves are depicted in Figure 1. 

Visual inspection of the divergence curves revealed different shapes for the different sensor locations. The divergence exponent λ_S_ was defined as the slope of the linear fit through the early phase of rapid divergence (thick lines Figure 1). Based on that, the interval for the linear fit was chosen at 0…25% of stride cycle for the sternum and foot and 0…20% of stride cycle for the sacrum and tibia. High values of λ_S_ reflect low running stability and vice versa. 

### 2.4. Statistical Analysis

Data analysis was performed using R (R Version 4.2.2 (2022), Statistical Computing, Vienna, Austria) through the library wrapper rpy2 for Python (rpy2, Version 3.4.5). To evaluate intra- and inter-day reliability, we calculated the two-way mixed effects absolute agreement intraclass correlation coefficients (ICC) for each sensor location and condition. ICC values were classified according to [31] as poor (ICC < 0.50), moderate (0.50 ≤ ICC < 0.75), good (0.75 ≤ ICC < 0.90), and excellent (ICC ≥ 0.90). We further computed absolute and relative bias and limits of agreement between repeated measurements. Relative values were normalized to the mean of the compared visits. 

To assess the agreement between different conditions, we used Bland–Altman statistics and calculated the absolute and relative bias and limits of agreement. Relative values were normalized to the mean over all visits of the reference condition. The reference condition for the comparisons was the treadmill (compared against both overground conditions) and the overground indoors condition (compared against overground outdoors). 

Additionally, to assess whether the differences between conditions were statistically significant, we performed two-way repeated measures analyses of variance (ANOVA). Within subject factors were running condition and session. When appropriate, Greenhouse–Geisser corrections were performed in case of violations of the sphericity assumption. Further, post hoc comparisons with the Bonferroni correction were performed if applicable. 

## 3. Results

Twelve of the twenty-five participants who participated in this study were female (48%). The average preferred running speed was 11.3 ± 2.1 km/h. An overview of participant demographics can be found in Table 1. 

Running stability values varied between sensor locations. Mean values and standard deviations over all three visits can be found in Table 2. 

### 3.1. Reliability

Reliability metrics for all sensor locations and conditions are listed in Table 2. Next, intra- and inter-day reliability are reported separately. 

#### 3.1.1. Intra-Day

We found moderate to excellent intra-day reliability values. The sacrum-mounted sensor in the treadmill condition exhibited the highest ICC (0.91; 95% CI = 0.81, 0.96). In contrast, we found the lowest intra-day reliability for the tibia-mounted sensor in the overground outdoor condition (0.55; 95% CI = 0.21, 0.78). Over all conditions, the sacrum showed the highest ICC values. Furthermore, ICC values for sternum and sacrum were rather consistent, whereas those of the tibia and foot sensors showed larger differences between conditions. 

Comparing measurements of the same day, the bias ranged between 7.4 to 8.8%, 0.3 to 0.5%, −1.0 to −0.4%, and 0.6 to 4.5% for sternum-, sacrum-, tibia-, and foot-mounted sensors, respectively. Limits of agreement varied from 11.9 to 38.8% over all sensors. 

#### 3.1.2. Inter-Day

When comparing measurements of consecutive days, ICC values indicated moderate-to-good inter-day reliability. Sternum sensor measurements in the outdoor overground condition revealed the highest inter-day reliability (0.90; 95% CI = 0.78, 0.95). We found the lowest values for overground outdoor measurements at the foot (0.66; 95% CI = 0.38, 0.84). On average, inter-day ICC values were higher than those of intra-day comparison.

Bias between repeated measurements ranged from −0.4 to 2.3%, 1.5 to 2.8%, −0.7 to 0.6%, and 1.3 to 2.2% for sternum-, sacrum-, tibia-, and foot-mounted sensors, respectively. Limits of agreement were lowest for the tibia sensor (8.9%) and highest for the sternum sensor (20.6%). 

### 3.2. Agreement between Conditions

Bland–Altman plots of the comparison of three conditions can be found in Figure 2. In addition, absolute and relative bias and limits of agreement are displayed in Table 3. For an easier comparison between sensor locations, we normalized bias and limits of agreement to sensor locations’ mean values. 

We found a negative bias between the treadmill and both overground conditions for all sensors. This indicates an increased running stability during the treadmill condition compared to the overground conditions. The largest mean difference between treadmill and overground running stability was present for the tibia sensor (bias and limits of agreement: indoor = −9.8 ± 15.6%; outdoor = −10.1 ± 14.1%). The best agreement to treadmill measurements was found for the sacrum sensor (bias and limits of agreement: indoor = −3.4 ± 13.1%; outdoor = −2.9 ± 12.6%).

A comparison of both overground conditions showed excellent agreement. Relative bias ranged from −0.3% to 0.5% over all sensor locations. The lowest and highest relative limits of agreement were present for the tibia and foot sensors, respectively. 

The ANOVA revealed significant differences between running conditions for all sensor locations (sternum: *p* < 0.001, η²_partial_ = 0.314; sacrum: *p* < 0.001, η²_partial_ = 0.280; tibia: *p* < 0.001, η²_partial_ = 0.680; foot: *p* < 0.001, η²_partial_ = 0.547). Post hoc comparisons indicated a significant difference between treadmill and overground indoor conditions for all sensors (sternum: *p* = 0.004; sacrum: *p* = 0.004; tibia: *p* < 0.001; foot: *p* < 0.001). Moreover, sternum (*p* = 0.004), sacrum (*p* = 0.023), tibia (*p* < 0.001), and foot (*p* < 0.001) sensors also showed significantly different values between treadmill and outdoor overground conditions. We did not find any statistically significant differences between the two overground conditions (sternum: *p* = 1.000; sacrum: *p* = 0.992; tibia: *p* = 1.000; foot: *p* = 1.000).

## 4. Discussion

The study’s main aim was the assessment of the intra- and inter-day reliability of running stability during treadmill and both indoor and outdoor overground running. Our results revealed moderate-to-excellent reliability over all conditions at four different body locations. On average, measuring running stability at the sacrum showed the highest intra- and inter-day reliabilities (ICC > 0.85). The lowest reliability was found for the tibia-mounted sensor, which was still moderate (ICC = 0.55). 

Ekizos et al. [17] reported the intra- and inter-day reliability of running stability during treadmill running. The average running speed was comparable to our study. They determined running stability based on vertical spine movement utilizing opto-electrical measurements. Their results mainly showed good reliability within and between days (ICC = 0.688–0.870). These findings are comparable to those of the sternum sensor during the treadmill condition in the present study (ICC = 0.62–0.85). Interestingly, the sacrum sensor displayed even higher reliability under almost all conditions. In contrast to this, tibia- and foot-mounted sensors exhibited lower reliability for almost all cases. It appears that more degrees of freedom between the measured body segment and the body’s center of mass can lead to more inconsistent running stability within participants. The position control of the body’s center of mass relative to the base of support (i.e., foot placement) is the main criterion for stable locomotion [32]. Thus, repeatedly maintaining high stability in the region of the sacrum might be of higher priority compared to more distal body segments. However, these findings are not to suggest measuring running stability at the sacrum only. Hamacher and co-workers found higher intra- and inter-day reliability in level-grounded *walking* for a trunk-mounted sensor compared to one mounted at the foot [21]. However, the lower reliability of the foot-mounted sensor was associated with a higher effect size (Hedges’ g = 1.18) than the trunk-mounted sensor (Hedges’ g = 0.51) when comparing the local dynamic stability of old and young, healthy adults. This finding coincides with the findings of Hoenig et al. [16]. They were able to distinguish between recreational and competitive runners based on a foot-mounted sensor but not on thorax- and sacrum-mounted sensors. Thus, placing sensors in more distal regions might be advantageous to detect meaningful differences in a tradeoff for slightly lower but still sufficient reliability. 

Running stability has been reported to be 2–4% lower in barefoot compared to shod running [13,14]. Further, recreational runners were shown to have an approximately 4% lower running stability compared to competitive runners [16]. The same study reported an increase of running stability over the course of a maximum effort run of below 4%. The bias between repeated measurements demonstrated in this study was below this for almost all sensors and conditions. Thus, we conclude that it is possible to detect small changes in running stability reliably at different body locations in different running environments. 

Secondly, this study aimed to evaluate the agreement between the different running conditions. We could demonstrate a systematic negative bias between the treadmill and both overground conditions for all sensors (Figure 2; Table 3). These reduced values of λ_S_ on the treadmill indicate a higher running stability. This confirms findings comparing local dynamic stability in treadmill and overground walking [24]. It has been hypothesized that the constant speed of the treadmill belt and the more constrained environment artificially increased stability [1,24,33]. We attempted to reduce the variation in running speed during the overground conditions by giving timing signals over the 50 m track. Still, we only measured the average speed over the distance. Instantaneous fluctuation in running speed may have still occurred and been one reason for the systematic bias.

Except for the sacrum sensor, intra-day reliability was lower in the overground outdoor compared to the overground indoor condition. In addition, only two out of four sensors showed good inter-day reliability at the overground outdoors condition (two moderate). During overground indoor running, all four sensor locations revealed good reliabilities (Table 2). This result may have been caused by an adaptation of the control structures to varying environmental conditions when running outdoors. However, since we did not measure perceptual (e.g., light conditions, pedestrians count, etc.), weather, or surface variables, this remains speculative. The effects are averaged out when comparing values of all three visits for the Bland–Altman bias and limits of agreement. 

Santuz et al. showed reduced local dynamic stability during running on uneven compared to even surfaces on a treadmill [12]. However, no significant differences in local dynamic stability of walking overground on different compliant surfaces were found by Chang et al. [34]. Hence, the influence of changes between typical running surfaces (e.g., tarmac, gravel, tartan) still needs to be investigated. In the current study, we did not find significant differences in running stability between the indoor and outdoor conditions with different surfaces (indoor: even, firm carpet; outdoor: even, concrete). 

With our approach to compare indoor and outdoor running with treadmill running, we aimed to shed light on the ecological validity of running stability measured under laboratory conditions. However, 50 m shuttle runs with a predefined running speed arguably do not reflect the real-world conditions of runners. Furthermore, our data were assessed by one researcher only. Inter-rater reliability is usually good for inertial sensor measurements of walking gait [34,35]. However, we could not confirm this for running stability with our study design. 

Despite these limitations, this is the first study demonstrating the reliability of running stability under different running conditions. Our findings provide an important foundation for future research of inertial sensor-based running stability measurements. Most inertial sensor-based running biomechanics research is conducted in laboratory settings [36]. We could demonstrate that outdoor measurements are almost equally reliable compared to controlled laboratory conditions. Small and low-cost wearable inertial sensors can be worn by runners throughout their regular training and competition runs. This allows for long-term monitoring of running stability, which in turn might provide valuable insights, e.g., into the development of running-related injuries.

## 5. Conclusions

In this study, we evaluated the reliability of running stability measured at four different body locations at three different running conditions. Over all locations and conditions, we could demonstrate moderate-to-excellent reliability. The highest reliability was found at the sacrum location. This lays an important foundation for future research on running stability. Caution is advised, however, when comparing values of different locations and between overground and treadmill conditions. Wearable inertial sensor-based measurements enable researchers to conduct data acquisition in the field. With our findings, we further encourage additional research on running stability under real-world conditions. This has the potential to elucidate new insights into runners’ movement abilities under varying conditions. 

## Figures and Tables

**Figure 1 sensors-23-00347-f001:**
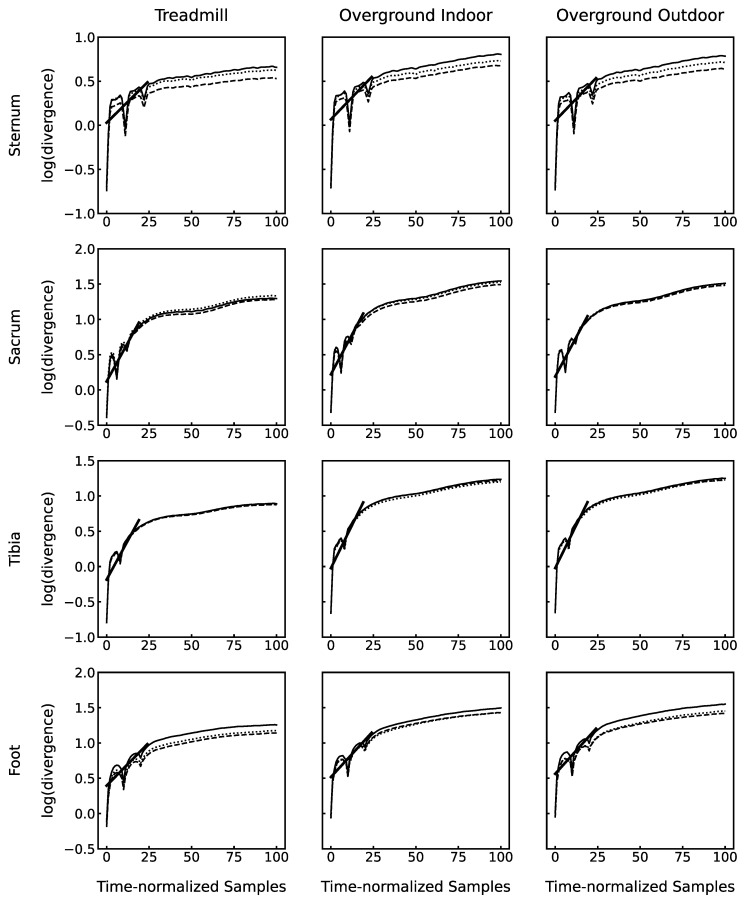
Average logarithmic divergence plots over all 25 participants for all conditions and sensor locations. Solid, dashed, and dotted lines represent data from the first, second, and third visits, respectively. Thick solid lines are the linear fit through the region of initial rapid divergence (exemplary for visit 1 only), representing running stability.

**Figure 2 sensors-23-00347-f002:**
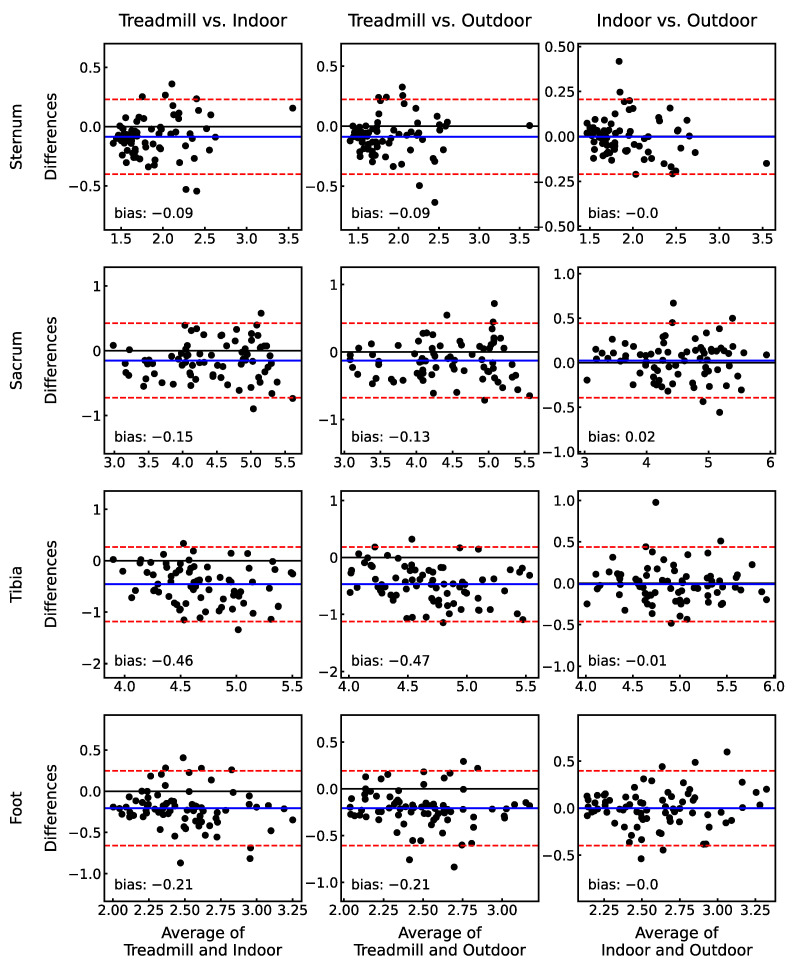
Bland–Altman plots for the agreement between the different conditions for all sensor locations. Solid green and dashed red lines represent bias and upper and lower limits of agreement, respectively. The condition mentioned first reflects the reference for the calculation.

**Table 1 sensors-23-00347-t001:** Participant demographics.

Demographic	Mean ± Standard Deviation
Number of Females/Males	12/13 (48%/52%)
Height (cm)	174.8 ± 8.4
Weight (kg)	69.4 ± 12.4
BMI (kg/m²)	22.6 ± 3.0
Age (years)	34.1 ± 10.0
Preferred running speed (km/h)	11.3 ± 2.1

**Table 2 sensors-23-00347-t002:** Results of the running stability measurements for each of the three visits, intra-, and inter-day reliability metrics. ICC, intraclass correlation coefficient; CI, confidence interval; LoA, Bland–Altman’s limits of agreement.

Condition	Sensor Location	Running Stability Mean (Standard Deviation)			Intra-Day Reliability			Inter-Day Reliability		
		Visit 1	Visit 2	Visit 3	ICC (95% CI)	Bias (95% CI)	LoA (95% CI)	ICC (95% CI)	Bias (95% CI)	LoA (95% CI)
Treadmill	Sternum	1.91 (0.53)	1.77 (0.34)	1.78 (0.32)	0.62 (0.32, 0.81)	0.14 (−0.02, 0.29)	0.73 (0.46, 0.99)	0.85 (0.69, 0.93)	0.04 (−0.04, 0.12)	0.38 (0.24, 0.52)
	Sacrum	4.39 (0.71)	4.38 (0.65)	4.28 (0.71)	0.91 (0.81, 0.96)	0.01 (−0.11, 0.13)	0.57 (0.36, 0.78)	0.89 (0.76, 0.95)	0.12 (−0.01, 0.25)	0.61 (0.39, 0.84)
	Tibia	4.43 (0.37)	4.45 (0.37)	4.46 (0.43)	0.63 (0.32, 0.82)	−0.02 (−0.15, 0.12)	0.63 (0.40, 0.86)	0.81 (0.61, 0.91)	−0.03 (−0.14, 0.08)	0.51 (0.32, 0.70)
	Foot	2.45 (0.27)	2.34 (0.27)	2.38 (0.32)	0.59 (0.26, 0.80)	0.11 (0.01, 0.20)	0.45 (0.29, 0.62)	0.72 (0.47, 0.87)	0.03 (−0.06, 0.12)	0.42 (0.27, 0.58)
Overground Indoor	Sternum	2.00 (0.47)	1.85 (0.32)	1.87 (0.32)	0.64 (0.32, 0.83)	0.15 (0.02, 0.29)	0.63 (0.40, 0.86)	0.86 (0.71, 0.93)	0.03 (−0.05, 0.10)	0.35 (0.22, 0.47)
	Sacrum	4.53 (0.75)	4.51 (0.63)	4.45 (0.64)	0.85 (0.69, 0.93)	0.02 (−0.13, 0.18)	0.75 (0.48, 1.02)	0.85 (0.70, 0.93)	0.07 (−0.08, 0.22)	0.71 (0.45, 0.97)
	Tibia	4.90 (0.41)	4.92 (0.48)	4.90 (0.50)	0.77 (0.55, 0.89)	−0.02 (−0.15, 0.10)	0.60 (0.38, 0.81)	0.89 (0.77, 0.95)	0.03 (−0.06, 0.13)	0.45 (0.28, 0.61)
	Foot	2.62 (0.35)	2.60 (0.34)	2.57 (0.31)	0.85 (0.69, 0.93)	0.02 (−0.06, 0.09)	0.37 (0.24, 0.51)	0.84 (0.68, 0.93)	0.05 (−0.02, 0.12)	0.35 (0.22, 0.47)
Overground Outdoor	Sternum	2.00 (0.50)	1.84 (0.33)	1.89 (0.35)	0.61 (0.28, 0.81)	0.17 (0.02, 0.31)	0.69 (0.44, 0.95)	0.90 (0.78, 0.95)	−0.01 (−0.07, 0.06)	0.32 (0.20, 0.43)
	Sacrum	4.50 (0.73)	4.48 (0.66)	4.44 (0.64)	0.91 (0.81, 0.96)	0.02 (−0.11, 0.14)	0.59 (0.38, 0.81)	0.86 (0.71, 0.94)	0.08 (−0.06, 0.22)	0.67 (0.43, 0.92)
	Tibia	4.91 (0.39)	4.96 (0.48)	4.89 (0.53)	0.55 (0.21, 0.78)	−0.05 (−0.22, 0.12)	0.82 (0.52, 1.12)	0.72 (0.46, 0.87)	0.03 (−0.14, 0.19)	0.77 (0.49, 1.05)
	Foot	2.67 (0.30)	2.56 (0.29)	2.57 (0.34)	0.72 (0.43, 0.87)	0.10 (0.02, 0.19)	0.40 (0.25, 0.54)	0.66 (0.38, 0.84)	0.06 (−0.05, 0.16)	0.50 (0.32, 0.68)

**Table 3 sensors-23-00347-t003:** Absolute and relative (to sensor location mean) bias and limits of agreement (LoA) with the 95% confidence intervals (CI) between the three conditions.

Sensor Location		Treadmill vs. Overground Indoor		Treadmill vs. Overground Outdoor		Overground Indoor vs. Overground Outdoor	
		Absolute	Relative	Absolute	Relative	Absolute	Relative
Sternum	Bias 95% CI	−0.09 (−0.12, −0.05)	−4.6% (−6.4%, −2.7%)	−0.09 (−0.13, −0.05)	−4.7% (−7.0%, −2.7%)	−0.00 (−0.03, 0.02)	−0.1% (−1.6%, 1.0%)
	LoA 95% CI	0.32 (0.25, 0.38)	17.0% (13.6%, 20.4%)	0.31 (0.25, 0.38)	16.8% (13.4%, 20.1%)	0.21 (0.17, 0.25)	10.9% (8.7%, 13.1%)
Sacrum	Bias 95% CI	−0.15 (−0.22, −0.08)	−3.4% (−5.0%, −1.8%)	−0.13 (−0.19, −0.06)	−2.9% (−4.3%, −1.4%)	0.02 (−0.03, 0.07)	0.5% (−0.7%, 1.6%)
	LoA 95% CI	0.58 (0.46, 0.70)	13.1% (10.5%, 15.7%)	0.56 (0.44, 0.67)	12.6% (10.1%, 15.1%)	0.42 (0.34, 0.51)	9.4% (7.5%, 11.3%)
Tibia	Bias 95% CI	−0.46 (−0.54, −0.37)	−9.8% (−11.5%, −7.9%)	−0.47 (−0.55, −0.39)	−10.1% (−11.7%, −8.3%)	−0.01 (−0.07, 0.04)	−0.3% (−1.4%, 0.8%)
	LoA 95% CI	0.73 (0.58, 0.88)	15.6% (12.5%, 18.8%)	0.66 (0.53, 0.79)	14.1% (11.2%, 16.9%)	0.45 (0.36, 0.54)	9.2% (7.4%, 11.1%)
Foot	Bias 95% CI	−0.21 (−0.26, −0.15)	−8.3% (−10.4%, −6.0%)	−0.21 (−0.25, −0.16)	−8.3% (−10.0%, −6.4%)	−0.00 (−0.05, 0.05)	−0.0% (−1.9%, 1.9%)
	LoA 95% CI	0.46 (0.36, 0.55)	18.3% (14.6%, 21.9%)	0.40 (0.32, 0.48)	16.1% (12.9%, 19.3%)	0.40 (0.32, 0.48)	15.4% (12.3%, 18.5%)

## Data Availability

The data presented in this study are available upon request from the corresponding author.

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
