# Peer review of "Reliability of Running Stability during Treadmill and Overground Running"

_sensors, 2022, doi:10.3390/s23010347_

Round 1

Reviewer 1 Report

The article addresses an important topic for runners, namely the running stability during treadmill and overground.

I have several comments.

Introduction. Be more precise when referring to reliability between days.

Participants. Clarify what word of mouth refers to. Are there other exclusion criteria (for example, neurologic or rheumatic diseases)?

Data collection. Twice on the first...The verb is missing.

Results. Table 1. I would recommend to have both genders includes. Otherwise, it seems like height, weight, etc. are referring to women only.

Were there any comparisons among different age groups?

Reviewer 2 Report

1-      Performing tests on different days and time would be influenced by circadian rhythms. Please explain how did you control for it?

2-      The manuscript requires more information on how stability is defined and why LyE is a measure of stability. What does lower and greater values mean?

3-      Please justify the reasoning behind sensor locations, especially for tibia and foot, are they on the dominant limb?

4-      Please add hypothesis in the aim paragraph

5-      Please clearly indicate what does regions 0…25% means?

6-      Number of digits should be similar for table 2, right now some of them have 3 and some 4 digits.

7-      It is useful to perform a statistical test to compare differences in running stability between different conditions and discuss the results.

8-      Please report the 95% CI of mean difference and the agreement limits to describe possible error of estimation due to sampling error.

9-      For the below statement you can add the citation:

“It has been hypothesized that the constant speed of the treadmill belt and the more constrained environment artificially increased stability.”

Fallahtafti et al., Margin of Stability May Be Larger and Less Variable during Treadmill Walking Versus Overground. Biomechanics
